**Data Availability Statement:** "The data used in this study are the property of the United States government Department of Veterans Affairs. Electronic health records contain substantial

# Variation in benefit among patients with serious mental illness who receive integrated psychiatric and primary care

Alexander S. Young[1,2,3]*, Jessica Skela[2], Evelyn T. Chang[3,4,5], Rebecca Oberman[3], Prabha Siddarth[2]

1 Desert Pacific Mental Illness Research Education and Clinical Center, Greater Los Angeles Veterans Healthcare System, Los Angeles, California, United States of America, 2 Department of Psychiatry, School of Medicine, University of California Los Angeles, Los Angeles, California, United States of America, 3 HSR&D Center for the Study of Healthcare Innovation, Implementation and Policy, Greater Los Angeles Veterans Healthcare System, Los Angeles, California, United States of America, 4 Department of Medicine, Veterans Affairs Greater Los Angeles Healthcare System, Los Angeles, California, United States of America, 5 Department of Medicine, David Geffen School of Medicine, University of California Los Angeles, Los Angeles, California, United States of America

* ayoung@ucla.edu

## Abstract

### Purpose

The population with serious mental illness has high risk for hospitalization or death due to unhealthy behaviors and inadequate medical care, though the level of risk varies substantially. Programs that integrate medical and psychiatric services improve outcomes but are challenging to implement and access is limited. It would be useful to know whether benefits are confined to patients with specific levels of risk.

### Methods

In a population with serious mental illness and increased risk for hospitalization or death, a specialized medical home integrated services and improved treatment and outcomes. Treatment quality, chronic illness care, care experience, symptoms, and quality of life were assessed for a median of 385 days. Analyses examine whether improvements varied by baseline level of patient risk.

### Results

Patients with greater risk were more likely to be older, more cognitively impaired, and have worse mental health. Integrated services increased appropriate screening for body mass index, lipids, and glucose, but increases did not differ significantly by level of risk. Integrated services also improved chronic illness care, care experience, mental health-related quality of life, and psychotic symptoms. There were also no significant differences by risk level.

sensitive information, private information and information that is protected by United States statute and regulation, including HIPAA. Therefore, it is not possible for us to share the data used in these analyses. Data can be requested from the authors or the research service of the Great Los Angeles Veterans Healthcare System, which can be reached at 1-310-268-4437 or VHAWLAResearchSvc@va.gov".

**Funding:** This research was supported by the U.S. Department of Veterans Affairs, Veterans Health Administration, Health Services Research and Development Service Quality Enhancement Research Initiative (SDP 12-177; AY), and Desert Pacific Mental Illness Research, Education and Clinical Center (MIRECC; AY). https://www.queri. research.va.gov/ Funders had no role in study design, data collection and analysis, decision to publish, or preparation of the manuscript.

**Competing interests:** The authors have declared that no competing interests exist.

## Conclusions

There were benefits from integration of primary care and psychiatric care at all levels of increased risk, including those with extremely high risk above the 95th percentile. When developing integrated care programs, patients should be considered at all levels of risk, not only those who are the healthiest.

## Introduction

Healthcare organizations are increasingly responsible for providing comprehensive care of populations of patients, and for improving treatment quality while controlling costs. Serious mental illnesses such as bipolar disorder or schizophrenia are present in a substantial proportion of patients with preventable emergency visits, hospitalizations or death [1,2]. A major cause of increased risk is inadequate primary care for common medical conditions [3–6]. There are interventions that integrate psychiatric and primary care and improve outcomes, but these require additional resources and reorganization of care, and access to these services is limited [7–10]. Patients with high risk for poor outcomes, including those with serious mental illness (SMI) have the most potential to benefit from these interventions. However, it is not clear whether care integration efforts benefit all patients with increased health risk, or whether these services can be focused more narrowly.

At a large Veterans Health Administration (VHA) healthcare system, psychiatric care was integrated into a primary care medical home that focused on patients with SMI and increased risk. The VHA refers to medical homes as Patient Aligned Care Teams (PACTs), and this medical home was referred to as SMI PACT [11]. In a prospective controlled trial, SMI PACT improved treatment experience, quality of care, and patient outcomes [8]. The program selected eligible patients for enrollment from individuals with increased risk. Risk was quantified using the Care Assessment Need (CAN) score, which the VHA calculates on a weekly basis for all patients [12]. The CAN score accurately estimates the risk of hospitalization or death using a model that includes many variables drawn from the domains of demographics, diagnoses, vital signs, medications, laboratory values and service utilization. Expressed as a percentile of risk, the CAN one-year risk for hospitalization or death increases from about 15% at the 75th percentile to 51% for all who are above the 95th percentile of risk [12].

Organizations have used risk prediction to focus enhanced service delivery on patients with high risk for poor outcomes [13–16]. Patients with greater risk would be expected to have the potential for greater improvement with enhanced programs or new interventions. However, they might also be more challenging to engage and treat, and very high risk may not be modifiable. There has been little understanding of how baseline risk contributes to improvement in treatment and outcomes with integration of care. It would be useful to know whether all patients benefit, or whether enhanced programs can be focused more narrowly. To inform future efforts to target service delivery, this manuscript examines whether improvements in treatment and outcomes vary by patients' level of risk.

## Methods

### Design

The SMI PACT project was registered with ClinicalTrials.gov (identifier NCT01668355) and the research methods and clinical trial results have been published previously [8,11,17]. Service

integration occurred at the Veterans Affairs Greater Los Angeles Healthcare System, one of the largest VHA healthcare systems in the United States. Patients were eligible if they had: 1) a CAN score over the 75[th] percentile of risk, and 2) SMI, which was defined as having a diagnosis of schizophrenia, schizoaffective disorder, bipolar disorder, recurrent major depression with psychosis, or chronic severe post-traumatic stress disorder requiring antipsychotic medication treatment. From 1,322 eligible patients, a sample of 164 were randomly selected and offered care from the SMI PACT medical home. This sample size was determined using power analysis regarding the number required to detect change in treatment utilization. Patients were recruited between May 3, 2016, and February 21, 2018. This study was approved by the IRB of the Veterans Affairs Greater Los Angeles Healthcare System. Participants provided written informed consent.

Data are drawn from in-person assessments that were performed by trained research staff who were blind to patients' CAN scores. Measures include gender, race, ethnicity, education attainment, psychiatric diagnosis, level of mental health recovery (MORS) [18], cognitive functioning in the domains of verbal learning and speed of processing (HVLT-R and digit symbol substitution test) [19,20], psychotic symptom severity (BASIS-R psychosis scale) [21], ambulatory care experience (ACES) [22], receipt of chronic illness care (PACIC) [23], patient activation (PAM-13) [24], and health related mental (MCS) and physical (PCS) components of quality of life (VR-12) [25]. VHA administrative information systems provided data on lab test results, height, weight, blood pressure, diagnoses, prescriptions, and services. These data were used to calculate Body Mass Index (BMI), and measures of treatment appropriateness and quality, including measures of metabolic screening and monitoring based on specifications from NCQA, HEDIS, and VA.

## Analysis

Patients were grouped into risk strata based on CAN percentile score at baseline. Demographics and medical record data were compared among risk groups using chi square tests for categorical variables and analysis of variance for continuous variables. These were univariate tests (one variable at a time). A cumulative logistic regression model was used to estimate the effect of individual patient characteristics on health risk. Odds ratios were calculated to assess the contribution of each variable associated with health risk. Goodness of fit was determined by calculating the area under the receiver operating characteristic curve. To examine the effect of health risk on treatment quality and outcomes, linear mixed effects models with repeated measures were estimated. Separate models were run for each of the continuous outcomes, with CAN group, time (baseline and post-intervention), and group x time interaction as predictors. Generalized linear mixed effects models were used for the binary outcomes (metabolic screening measures), with the same predictors of CAN group, time, and group x time interaction. Statistical significance was defined as a 2-sided p-value < .05.

## Results

Table 1 describes the sample, including characteristics and in-person assessment measures stratified by CAN risk percentile of 75–85 (n = 89), 85–95 (n = 51) and 95–99 (n = 23). Nearly 80% of patients had some college education, 35% had schizophrenia, 35% had bipolar disorder, and 28% had chronic, severe, disabling Post-Traumatic Stress Disorder. Cognitive functioning and health-related quality of life were substantially below general population means (mean HVLT-R = 5.1, SD = 1.8; mean Digit Symbol Test = 38.4, SD = 10.9; mean PCS = 36.4, SD = 11.1; mean MCS = 39.9, SD = 13.1). Greater health risk was significantly associated with advancing age (odds ratio per decade = 1.49, p = 0.02) and worse mental health-related quality

**Table 1. The characteristics of patients stratified by level of health risk.**

| Characteristic | Overall (N = 164) | CAN 75–85 (N = 89) | CAN 85–95 (N = 51) | CAN 95–99 (N = 23) | |
|---|---|---|---|---|---|
| | N (%) or Mean±SD | N (%) or Mean±SD | N (%) or Mean±SD | N (%) or Mean±SD | p-value |
| Gender | | | | | 0.93 |
| Female | 14 (8.5) | 7 (7.9) | 5 (9.8) | 2 (8.7) | |
| Male | 150 (91.5) | 82 (92.1) | 46 (90.2) | 21 (91.3) | |
| Age, years | 58.7 ± 10.4 | 57.3 ± 11.1 | 59.7 ± 8.7 | 63.2 ± 7.7 | 0.01 |
| Race | | | | | 0.63 |
| White | 59 (36.2) | 35 (39.8) | 16 (31.4) | 8 (34.8) | |
| Black | 74 (45.4) | 38 (43.2) | 27 (52.9) | 9 (39.1) | |
| Other | 30 (18.4) | 15 (17.0) | 8 (15.7) | 6 (26.1) | |
| Ethnicity | | | | | 0.89 |
| Hispanic or Latino | 22 (13.7) | 11 (12.6) | 7 (14.3) | 3 (13.0) | |
| Not Hispanic or Latino | 138 (86.3) | 76 (87.4) | 42 (85.7) | 20 (87.0) | |
| Education | | | | | 0.76 |
| Less than high school | 7 (4.3) | 3 (3.4) | 3 (5.9) | 1 (4.3) | |
| High school or equivalent | 26 (15.9) | 18 (20.2) | 5 (9.8) | 3 (13.0) | |
| Some college | 115 (70.1) | 60 (67.4) | 38 (74.5) | 16 (69.7) | |
| Some graduate school | 16 (9.7) | 8 (9.0) | 5 (9.8) | 3 (13.0) | |
| Diagnosis | | | | | 0.40 |
| Schizophrenia | 57 (34.8) | 28 (31.5) | 20 (39.2) | 9 (39.1) | |
| Bipolar disorder | 57 (34.8) | 34 (38.2) | 18 (35.3) | 4 (17.4) | |
| Major depression with psychosis | 4 (2.4) | 1 (1.1) | 2 (3.9) | 1 (4.3) | |
| Chronic disabling PTSD | 46 (28.0) | 26 (29.2) | 11 (21.6) | 9 (39.1) | |
| Recovery, MORS | 6.4 ± 0.6 | 6.4 ± 0.6 | 6.3 ± 0.6 | 6.4 ± 0.7 | 0.52 |
| Cognition, HVLT-R | 5.1 ± 1.8 | 5.2 ± 1.7 | 5.3 ± 1.7 | 4.3 ± 1.9 | 0.12 |
| Cognition, Digit Symbol | 38.4 ± 10.9 | 39.5 ± 10.8 | 38.6 ± 11.1 | 33.1 ± 9.5 | 0.03 |
| Chronic illness care | 2.7 ± 1.0 | 2.6 ± 1.0 | 2.82 ± 1.1 | 2.54 ± 1.0 | 0.97 |
| Ambulatory care experience | 68.1 ± 20.5 | 66.8 ± 18.8 | 76.7 ± 20.3 | 57. ± 25.6 | 0.85 |
| Patient activation | 2.1 ± 0.4 | 2.2 ± 0.4 | 2.1 ± 0.5 | 1.9 ± 0.4 | 0.07 |
| Physical quality of life | 36.4 ± 11.1 | 37.7 ± 12.0 | 33.5 ± 9.5 | 36.8 ± 9.5 | 0.26 |
| Mental health quality of life | 39.9 ± 13.1 | 41.8 ± 13.0 | 39.6 ± 12.8 | 34.3 ± 12.7 | 0.02 |
| Psychotic symptoms | 0.9 ± 0.8 | 0.8 ± 0.7 | 1.0 ± 0.9 | 0.9 ± 0.8 | 0.20 |

of life (odds ratio = 0.93, p < .001). Risk was not significantly associated with gender, race, ethnicity, educational level, specific psychiatric diagnosis, cognition, psychiatric symptoms, physical health related quality of life, chronic illness care, experience of care, or patient activation. Diagnostic accuracy of this model was 72.3% (95% CI 65.8–78.8).

Participants who were assessed at follow-up (n = 133) participated in the intervention for a median of 385 days (mean = 427, SD = 106). Mean days of participation were 414 (SD = 97.9) for the 75–85 CAN group, 446 (SD = 116.9) for the 85–95 CAN group, and 441 (SD = 110.4) for the 95–99 CAN group. These three groups were not significantly different in their number of days of participation (Kruskal Wallis chi square (2) = 5.4, p = .07).

As previously reported, the SMI PACT intervention improved chronic illness care, care experience, mental health-related quality of life and psychotic symptoms, as well as screening for body mass index, lipids and glucose, compared to usual care [8]. Table 2 presents improvements in the quality of care with SMI PACT, stratified by risk group. There were improvements in the appropriate screenings for body mass index, lipids, and glucose, however, these did not vary by stratum of risk (Wald $\chi^2$ = 0.0–1.40, p = 0.23–1).

**Table 2. The effect of integrated care on treatment quality stratified by baseline risk.**

| Measure | Before | After | β (CAN group | Chi-Square | p-value |
|---|---|---|---|---|---|
| | Medical Home | Medical Home | x time), p-value | (df) | |
| | n (%) | n (%) | | | |
| BMI Screening | | | | | |
| CAN Group 75–85 | 69 (77.5) | 75 (84.3) | | 0.18 (1) | 0.7 |
| CAN Group 85–95 | 42 (82.4) | 46 (90.2) | 0.239, 0.7 | | |
| CAN Group 95–99 | 21 (91.3) | 22 (95.7) | 0.299, 0.8 | | |
| Blood Pressure Screening | | | | | |
| CAN Group 75–85 | 72 (80.9) | 74 (83.1) | | 0 (1) | 1 |
| CAN Group 85–95 | 48 (94.1) | 48 (94.1) | -0.153, 0.9 | | |
| CAN Group 95–99 | 23 (100.0) | 23 (100.0) | -0.153, 0.9 | | |
| Lipid Screening | | | | | |
| CAN Group 75–85 | 33 (37.1) | 42 (47.2) | | 1.4 (1) | 0.2 |
| CAN Group 85–95 | 12 (23.5) | 31 (60.8) | 1.201, 0.03 | | |
| CAN Group 95–99 | 7 (30.4) | 15 (65.2) | 1.034, 0.14 | | |
| Glucose Screening | | | | | |
| CAN Group 75–85 | 46 (51.7) | 53 (59.6) | | 0.52 (1) | 0.5 |
| CAN Group 85–95 | 35 (68.6) | 44 (86.3) | 0.736, 0.2 | | |
| CAN Group 95–99 | 19 (82.6) | 21 (91.3) | 0.474. 0.6 | | |

Table 3 presents improvements in patient outcomes, stratified by risk group. While there were improvements in chronic illness care, care experience, mental health related quality of life, and psychotic symptoms, improvements also did not vary significantly by risk (F = 0.21–2.86, p = 0.06–0.8). Even the stratum with extremely high risk (> 95th percentile) improved in chronic illness care and physical health related quality of life.

## Discussion

This study suggests that improvements in care and outcomes do not vary significantly by level of health risk. There is improvement in healthcare and outcomes across multiple domains even for those with extremely high risk. However, the magnitude of improvement can vary. With this study's intervention, there was minimal improvement in patient activation or physical health-related quality of life. While there was substantial improvement in the other outcomes at each level of risk, psychotic symptoms did not improve for the group with the highest level of risk. While not all people with SMI have increased health risk, there is a large population at high risk of hospitalization or death. Using risk prediction, this project identified the population of patients with SMI and high risk for hospitalization or death. Patients with greater health risk were more likely to be older, more cognitively impaired, and have worse mental health. There were benefits from integration of primary care and psychiatric care at all levels of increased risk, including those with risk above the 95th percentile. The patients with SMI were very receptive to medical preventive services [26], and substantial improvements were seen in some measures of screening, such as for lipids.

A potential limitation of this study is that it was conducted in the VHA. The VHA offers comprehensive healthcare services and provides people with SMI better access to medical care than some other organizations. The health risk of people with SMI could be greater in systems with more problems in access to care. Also, while the sample size in this study was based on an a priori power calculation, it is possible that we did not have enough power to detect small differences in outcomes among the three risk groups.

**Table 3. The effect of integrated care on patient outcomes stratified by baseline risk.**

| Measure | Before | After | Change | Statistic | |
|---|---|---|---|---|---|
| | Medical Home | Medical Home | | | |
| | mean (SD) | mean (SD) | mean (SD), p-value | F (ndf,ddf) | p-value |
| Chronic illness care | | | | 1.9 (2, 115) | 0.15 |
| CAN 75–85 | 2.5 (0.9) | 3.6 (1.0) | 0.98 (1.2), < 0.001 | | |
| CAN 85–95 | 2.8 (1.0) | 3.5 (0.8) | 0.69 (0.9), < 0.001 | | |
| CAN 95–99 | 2.4 (1.0) | 2.9 (0.9) | 0.66 (0.9), 0.05 | | |
| Care experience | | | | 0.32 (2, 138) | 0.7 |
| CAN 75–85 | 70.3 (23.8) | 85.3 (17.4) | 13.25 (22.9), <0.001 | | |
| CAN 85–95 | 76.7 (22.2) | 87.0 (13.1) | 11.07 (23.6), 0.01 | | |
| CAN 95–99 | 69.3 (26.6) | 81.4 (21.0) | 15.15 (28.3), 0.02 | | |
| Patient activation | | | | 0.56 (2, 139) | 0.6 |
| CAN 75–85 | 2.2 (0.4) | 2.2 (0.5) | 0.05 (0.4), 0.3 | | |
| CAN 85–95 | 2.1 (0.5) | 2.3 (0.5) | 0.11 (0.4), 0.06 | | |
| CAN 95–99 | 1.90 (0.4) | 1.9 (0.4) | 0.02 (0.3), 0.8 | | |
| Physical health-related quality of life | | | | 2.86 (2, 138) | 0.06 |
| CAN 75–85 | 37.7 (12.0) | 39.2 (11.5) | 0.77 (9.2), 0.3 | | |
| CAN 85–95 | 33.5 (9.5) | 34.9 (9.1) | 0.50 (9.1), 0.6 | | |
| CAN 95–99 | 36.8 (9.5) | 32.0 (9.2) | -3.97 (9.3), 0.04 | | |
| Mental health-related quality of life | | | | 0.21 (2, 140) | 0.8 |
| CAN 75–85 | 41.8 (13.0) | 44.2 (12.6) | 2.37 (12.1), 0.08 | | |
| CAN 85–95 | 39.5 (12.8) | 43.1 (12.5) | 3.18 (10.5), 0.07 | | |
| CAN 95–99 | 34.3 (12.7) | 38.0 (11.9) | 4.38 (13.6), 0.12 | | |
| Psychotic symptoms | | | | 0.36 (2,140) | 0.7 |
| CAN 75–85 | 0.8 (0.7) | 0.6 (0.7) | -0.15 (0.7), 0.04 | | |
| CAN 85–95 | 1.0 (0.9) | 0.8 (0.8) | -0.24 (0.8), 0.05 | | |
| CAN 95–99 | 0.9 (0.8) | 0.9 (0.8) | -0.09 (0.7), 0.7 | | |

Studies have found that health can be improved in populations with SMI with interventions that integrate medical and psychiatric care [27–29]. The large disparities in health and healthcare seen in this population can be improved. The routine use of health risk assessment can support the implementation and conduct of interventions to improve medical care of people with increased health risk and can strengthen such models. In the context of limited resources, it is reasonable to focus integration of care on patients with increased risk for poor outcomes since there is the most potential benefit in these patients. Within this population, there is similar benefit for patients at each level of increased risk, including those with extremely high risk for poor outcomes. Integrated care programs should consider recruiting all patients with health risk, not only those who are the heathiest.

## Acknowledgments

The contents of this publication and the views expressed herein do not necessarily represent the views of the Department of Veterans Affairs or affiliated institutions.

## Author Contributions

**Conceptualization:** Alexander S. Young, Evelyn T. Chang.

**Data curation:** Alexander S. Young, Jessica Skela, Prabha Siddarth.

**Formal analysis:** Jessica Skela, Prabha Siddarth.

**Funding acquisition:** Alexander S. Young.

**Investigation:** Alexander S. Young, Evelyn T. Chang, Rebecca Oberman.

**Methodology:** Alexander S. Young.

**Project administration:** Alexander S. Young, Rebecca Oberman.

**Resources:** Alexander S. Young.

**Supervision:** Alexander S. Young, Rebecca Oberman.

**Writing – original draft:** Alexander S. Young.

**Writing – review & editing:** Alexander S. Young, Jessica Skela, Evelyn T. Chang, Prabha Siddarth.

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
