## [Decision Letter · Decision Letter 0]

20 Mar 2024

PONE-D-24-02148Variation in Benefit among Patients with Serious Mental Illness Who Receive Integrated Psychiatric and Primary CarePLOS ONE

Dear Dr. Young,

Thank you for submitting your manuscript to PLOS ONE. After careful consideration, we feel that it has merit but does not fully meet PLOS ONE’s publication criteria as it currently stands. Therefore, we invite you to submit a revised version of the manuscript that addresses the points raised during the review process.

We look forward to receiving your revised manuscript.

Kind regards,

Mu-Hong Chen, M.D., Ph.D.

Academic Editor

PLOS ONE

Journal Requirements:

https://link.springer.com/article/10.1007/s11606-021-07270-x?

In your revision ensure you cite all your sources (including your own works), and quote or rephrase any duplicated text outside the methods section. Further consideration is dependent on these concerns being addressed.

This research was supported by the U.S. Department of Veterans Affairs, Veterans Health Administration, Health Services Research and Development Service Quality Enhancement Research Initiative (SDP 12-177; AY), and Desert Pacific Mental Illness Research, Education and Clinical Center (MIRECC; AY). 

https://www.queri.research.va.gov/

Funders had no role in study design, data collection and analysis, decision to publish, or preparation of the manuscript.

This research was supported by the U.S. Department of Veterans Affairs, Veterans Health Administration, Health Services Research and Development Service Quality Enhancement Research Initiative (SDP 12-177), and Desert Pacific Mental Illness Research, Education and Clinical Center (MIRECC). The contents of this publication and the views expressed herein do not necessarily represent the views of the Department of Veterans Affairs or affiliated institutions. 

This research was supported by the U.S. Department of Veterans Affairs, Veterans Health Administration, Health Services Research and Development Service Quality Enhancement Research Initiative (SDP 12-177; AY), and Desert Pacific Mental Illness Research, Education and Clinical Center (MIRECC; AY). 

https://www.queri.research.va.gov/

Funders had no role in study design, data collection and analysis, decision to publish, or preparation of the manuscript.

Reviewers' comments:

Reviewer's Responses to Questions

**Comments to the Author**

1. Is the manuscript technically sound, and do the data support the conclusions?

Reviewer #1: Yes

Reviewer #2: Partly

2. Has the statistical analysis been performed appropriately and rigorously? 

Reviewer #1: I Don't Know

Reviewer #2: No

3. Have the authors made all data underlying the findings in their manuscript fully available?

Reviewer #1: No

Reviewer #2: No

4. Is the manuscript presented in an intelligible fashion and written in standard English?

Reviewer #1: Yes

Reviewer #2: Yes

5. Review Comments to the Author

Reviewer #1: I) General comments

Authors present the results form a clinical trial on an integrated primary and psychiatric care home intervention for patients (n=164) with severe mental illness (SMI) and increased risk of hospitalisation or death. The authors report improvements across healthcare and patient outcomes across different levels of risk. Authors thus conclude that the intervention would benefit patients across different levels of risk, and not just those at the highest 95th percentile.

The topic of the manuscript is relevant, given the high risk of poor physical health outcomes among people with SMI, including premature death. Overall, the manuscript is correctly written and detailed enough to allow for replication. I add below my comments and suggestions for improvements of the manuscript.

II) Abstract:

- Authors indicate that the study was conducted “in a population with … increased risk”, but it is not explicitly mentioned what type of risk they refer to. While this is alluded to in the previous paragraph, I think it would better clarify the study’s methods if authors indicate that they refer to increased risk of hospitalisation or death.

- Authors indicate that the sample was followed for a median of 401 days, but this information is not mentioned later in the main text (Methods or Results sections).

III) Introduction: the introduction is well-structure, and clearly presents the problem and objectives of the study.

IV) Methods: overall, the methods section is well written, and the data collection and analysis strategy is detailed enough to allow for replication. A few comments:

- Page 4: There is a duplicated “random” in the final line: “a random sample of 164 were randomly selected”.

- Page 4: authors indicate that 164 participants were randomly selected, but no justification is given for the sample size. For example, was this sample size selected base on power analysis or due to practical reasons?

- More information would be helpful regarding the collection of outcome variables. Authors indicate that outcome data is drawn from in-person assessments (page 5). However, it is not mentioned whether the raters were blind to patient allocation to CAN score groups. Authors should also specify who conducted the in-person assessments. In particular, if the assessments were done by members of the research group, or by clinicians blind to the study’s objectives. This could introduce a risk of assessment bias that – if present - should also be acknowledged in the limitations.

V) Results: overall, the results are thoroughly and clearly presented. A few observations:

- Authors provide pre-intervention (“before medical home”) and a post-intervention (“after medical home”) outcomes scores, but no information is given on the time between each time measure. This should be better clarified to help interpret the results.

- Asterisk in titles of Tables 2-3 are not explained.

- Table 2: are the values in the “change” column significant? This is not indicated, as in Table 3.

VI) Discussion: overall, the discussion in balanced and the conclusions mostly derive from the study’s results. However, the section would benefit from a more nuanced discussion on a few key points:

- Authors indicate that there were improvements in outcomes across all risk groups. Authors mention “lipids screening” as a healthcare outcome that shows an improvement. However, no mention is given to the other treatment quality outcomes, particularly those that do not seem to show a significant improvement (e.g. BMI screening and blood pressure). Addressing this would provide a more balanced discussion.

- Similarly, the section would benefit from a more detailed discussion of the results related to “patient outcomes” (Table 3). In particular, some outcomes do not seem to show a significant improvement for some of the risk groups (e.g. psychotic symptoms for the CAN 95-99 group) or for any of the three risk groups (e.g. patient activation). This should be better acknowledged for a more accurate summary of the study’s results.

- The manuscript would benefit from a more explicit acknowledgement of the study’s strength and limitations. In particular, methodological aspects such as the risk of potential assessment bias raised above (if applicable).

- Citations should be provided for the statement in page 10: “Studies have found that health can be improved in populations with SMI with interventions that integrate medical and psychiatric care”.

Reviewer #2: The authors present results from a study of the impact of integrated services on health screening and patient outcomes, stratified by baseline patient risk. The study was conducted in a random sample of individuals from a large Veterans Affairs network. The authors find improvements pre-post integrated services, but no significant differences across risk groups. The manuscript will be strengthened if the authors consider the following points.

1. Authors report that median follow-up is 401 days, suggesting that patients were in the medical home for varying lengths of time. Did the time in the home differ between risk groups? It does not appear as though time in the home is accounted for in any way. Since the integrated services were delivered at the medical home, time in the home could have an impact on differences pre and post home. Authors need to address this difference in their analyses.

2. It is unclear if the p-values reported in Table 1 are from simple models (one independent variable associated with risk level) or from a joint model including all of the variables. This should be clarified both in the methods and in the table, so that readers know what they are looking at.

3. Were outcome measures of interest only collected before and after the medical home? The models fit for Tables 2 and 3 (as described in the Analysis section) seem to only utilize those two time points. If other assessments were available, why didn't the authors use that information, which would better capture potential fluctuation over time?

4. Table 2 gives the percentage receiving services before the medical home and after the medical home, as well as the difference in those numbers. Is it true that everyone who was receiving the services before the medical home was still receiving the services after the medical home? If not, "change" cannot be captured simply in the difference in number before and after the medical home.

5. Table 3 - are these observed means and standard deviations before and after the medical home? Was everyone seen both before and after the medical home? How is the "Change" column computed? The reason I ask is the mean change does not appear to correspond to the difference in means before and after (though it should if everyone is seen at both time points).

6. In the models for Table 3, did authors verify the assumptions? Some of the outcomes seem like they have the potential to be highly skewed, which often suggests underlying assumptions of the mixed effects models will be violated. Were any covariates included in the models, since there were observed differences between the risk groups at baseline?

7. Authors claim that improvements do not vary significantly by risk groups. While they did not find significant differences, did authors do a power calculation ahead of time to understand what level of difference they were powered to detect? A lack of significance does not prove there are no differences, so authors should be careful in how they interpret their findings.

Minor points:

1. Authors should clarify in the analysis section that patients were stratified based on baseline CAN percentile score, since earlier, they indicate that CAN scores are generated weekly.

2. Authors should also clarify that the area under the curve is area under the ROC curve, if that is what is being reported.

3. Table 1 - the numbers in the risk groups do not add to the overall column (there appears to be 1 person missing a bunch of data, but that is not reflected in the overall column). This discrepancy should be corrected.

4. Tables 2 and 3 have asterisks after "baseline risk" but do not define the asterisk.

6. PLOS authors have the option to publish the peer review history of their article (what does this mean?). If published, this will include your full peer review and any attached files.

Reviewer #1: No

Reviewer #2: No

---

## [Author Response · Author response to Decision Letter 0]

14 Apr 2024

Reviewer #1

II) Abstract

- Authors indicate that the study was conducted “in a population with … increased risk”, but it is not explicitly mentioned what type of risk they refer to. While this is alluded to in the previous paragraph, I think it would better clarify the study’s methods if authors indicate that they refer to increased risk of hospitalisation or death.

 We have clarified this as suggested.

- Authors indicate that the sample was followed for a median of 401 days, but this information is not mentioned later in the main text (Methods or Results sections).

 While the median duration of participation in the overall study was 401 days, for the intervention group studied in this manuscript, the median number of days to follow-up assessment was 385. We have clarified and corrected this, and also added this information to the first paragraph of the Methods.

IV) Methods

- Page 4: There is a duplicated “random” in the final line: “a random sample of 164 were randomly selected”.

 This is fixed.

- Page 4: authors indicate that 164 participants were randomly selected, but no justification is given for the sample size. For example, was this sample size selected base on power analysis or due to practical reasons?

 This was determined using a priori power analysis. This is now reported in the first paragraph of the Methods.

- More information would be helpful regarding the collection of outcome variables. Authors indicate that outcome data is drawn from in-person assessments (page 5). However, it is not mentioned whether the raters were blind to patient allocation to CAN score groups. Authors should also specify who conducted the in-person assessments. In particular, if the assessments were done by members of the research group, or by clinicians blind to the study’s objectives. This could introduce a risk of assessment bias that – if present - should also be acknowledged in the limitations.

 Assessments were performed by research staff who were blind to CAN score. This is now stated in the Methods.

V) Results

- Authors provide pre-intervention (“before medical home”) and a post-intervention (“after medical home”) outcomes scores, but no information is given on the time between each time measure. This should be better clarified to help interpret the results.

 We now present the number of days between each time measure at the end of the second paragraph of the Results section.

- Asterisk in titles of Tables 2-3 are not explained.

 The asterisks were an error and have been removed.

- Table 2: are the values in the “change” column significant? This is not indicated, as in Table 3.

 The change column in Table 2 was the difference between the number of participants evaluated before and after the intervention. As pointed out by the other reviewer, they are not informative. The mixed model estimates (betas and so on) are a better representation. Therefore, we have deleted the change column. Also, we have added p-values to the β estimates to indicate significance of change.

VI) Discussion

- Authors indicate that there were improvements in outcomes across all risk groups. Authors mention “lipids screening” as a healthcare outcome that shows an improvement. However, no mention is given to the other treatment quality outcomes, particularly those that do not seem to show a significant improvement (e.g. BMI screening and blood pressure). Addressing this would provide a more balanced discussion.

 We have added text to the Discussion first paragraph acknowledging that the magnitude of improvement varied by domain.

- Similarly, the section would benefit from a more detailed discussion of the results related to “patient outcomes” (Table 3). In particular, some outcomes do not seem to show a significant improvement for some of the risk groups (e.g. psychotic symptoms for the CAN 95-99 group) or for any of the three risk groups (e.g. patient activation). This should be better acknowledged for a more accurate summary of the study’s results.

 We have added 3 sentences in the first paragraph of the Discussion presenting these results, as suggested.

- The manuscript would benefit from a more explicit acknowledgement of the study’s strength and limitations. In particular, methodological aspects such as the risk of potential assessment bias raised above (if applicable).

 We have expanded the limitations section. There is not likely to have been assessment bias, as we discuss above.

- Citations should be provided for the statement in page 10: “Studies have found that health can be improved in populations with SMI with interventions that integrate medical and psychiatric care”.

We have added three citations for this statement.

Reviewer #2

1. Authors report that median follow-up is 401 days, suggesting that patients were in the medical home for varying lengths of time. Did the time in the home differ between risk groups? It does not appear as though time in the home is accounted for in any way. Since the integrated services were delivered at the medical home, time in the home could have an impact on differences pre and post home. Authors need to address this difference in their analyses.

 We have determined the days of participation for each risk group, and this is now presented in the second paragraph of the Results section as means with standard deviations. There were some numerical differences in the duration of participation by risk group. However, these differences were not linearly related to risk and were not significantly different with a Chi Square test. This is now presented in the Results section.

2. It is unclear if the p-values reported in Table 1 are from simple models (one independent variable associated with risk level) or from a joint model including all of the variables. This should be clarified both in the methods and in the table, so that readers know what they are looking at.

 The p-values are from univariate tests (one independent variable at a time). We now specify this in the Analysis section.

3. Were outcome measures of interest only collected before and after the medical home? The models fit for Tables 2 and 3 (as described in the Analysis section) seem to only utilize those two time points. If other assessments were available, why didn't the authors use that information, which would better capture potential fluctuation over time?

 Outcome measures of interest were only collected before and after participation in the medical home.

4. Table 2 gives the percentage receiving services before the medical home and after the medical home, as well as the difference in those numbers. Is it true that everyone who was receiving the services before the medical home was still receiving the services after the medical home? If not, “change” cannot be captured simply in the difference in number before and after the medical home.

 Yes, we agree. To improve presentation of results, we have removed the change column from Table 2, and have added statistics for Beta coefficients as discussed above.

5. Table 3 - are these observed means and standard deviations before and after the medical home? Was everyone seen both before and after the medical home? How is the "Change" column computed? The reason I ask is the mean change does not appear to correspond to the difference in means before and after (though it should if everyone is seen at both time points).

 Not all participants who were seen at baseline were assessed at follow-up. The means ‘before’ are computed on all participants assessed at baseline (n=163) and the means ‘after’ are computed on all participants assessed at follow-up (n=133).

6. In the models for Table 3, did authors verify the assumptions? Some of the outcomes seem like they have the potential to be highly skewed, which often suggests underlying assumptions of the mixed effects models will be violated. Were any covariates included in the models, since there were observed differences between the risk groups at baseline?

 We inspected the data for outliers, normality, and homogeneity of variance and we also examined the residual plots. No covariates were used in the models, since the CAN score itself incorporates many of the demographic and clinical characteristics (that are different between groups).

7. Authors claim that improvements do not vary significantly by risk groups. While they did not find significant differences, did authors do a power calculation ahead of time to understand what level of difference they were powered to detect? A lack of significance does not prove there are no differences, so authors should be careful in how they interpret their findings.

 The sample size was determined using a power analysis of the number of participants required to detect change in treatment utilization. We now describe this in the Methods section. While the sample size in this study was based on this power calculation, it is possible that we did not have enough power to detect small differences in outcomes among the three risk groups. We have added text stating this to the limitations section of the Discussion.

1. Authors should clarify in the analysis section that patients were stratified based on baseline CAN percentile score, since earlier, they indicate that CAN scores are generated weekly.

 We have added text on this to the Analysis section.

2. Authors should also clarify that the area under the curve is area under the ROC curve, if that is what is being reported.

 We have added text on this to the Analysis section.

3. Table 1 - the numbers in the risk groups do not add to the overall column (there appears to be 1 person missing a bunch of data, but that is not reflected in the overall column). This discrepancy should be corrected.

 This is due to the fact that one participant was missing a CAN score at baseline.

4. Tables 2 and 3 have asterisks after "baseline risk" but do not define the asterisk.

 The asterisks were an error and have been removed.

---

## [Decision Letter · Decision Letter 1]

10 May 2024

Variation in Benefit among Patients with Serious Mental Illness Who Receive Integrated Psychiatric and Primary Care

PONE-D-24-02148R1

Dear Dr. Alexander S. Young,

We’re pleased to inform you that your manuscript has been judged scientifically suitable for publication and will be formally accepted for publication once it meets all outstanding technical requirements.

Kind regards,

Mu-Hong Chen, M.D., Ph.D.

Academic Editor

PLOS ONE

Additional Editor Comments (optional):

Reviewers' comments:

Reviewer's Responses to Questions

**Comments to the Author**

1. If the authors have adequately addressed your comments raised in a previous round of review and you feel that this manuscript is now acceptable for publication, you may indicate that here to bypass the “Comments to the Author” section, enter your conflict of interest statement in the “Confidential to Editor” section, and submit your "Accept" recommendation.

Reviewer #1: All comments have been addressed

Reviewer #2: All comments have been addressed

2. Is the manuscript technically sound, and do the data support the conclusions?

Reviewer #1: Yes

Reviewer #2: (No Response)

3. Has the statistical analysis been performed appropriately and rigorously? 

Reviewer #1: N/A

Reviewer #2: (No Response)

4. Have the authors made all data underlying the findings in their manuscript fully available?

Reviewer #1: No

Reviewer #2: (No Response)

5. Is the manuscript presented in an intelligible fashion and written in standard English?

Reviewer #1: Yes

Reviewer #2: (No Response)

6. Review Comments to the Author

Reviewer #1: All my previous comments and observations have been addressed. I have not further comments or observations for this new version of the amnuscript.

Reviewer #2: (No Response)

7. PLOS authors have the option to publish the peer review history of their article (what does this mean?). If published, this will include your full peer review and any attached files.

Reviewer #1: No

Reviewer #2: No

---

## [Editor Report · Acceptance letter]

14 May 2024

PONE-D-24-02148R1 

PLOS ONE

Dear Dr. Young, 

I'm pleased to inform you that your manuscript has been deemed suitable for publication in PLOS ONE. Congratulations! Your manuscript is now being handed over to our production team.

Kind regards, 

on behalf of

Dr. Mu-Hong Chen 

Academic Editor

PLOS ONE